# Assessment of the technological viability of photoelectrochemical devices for oxygen and fuel production on Moon and Mars

Byron Ross[1], Sophia Haussener [2] & Katharina Brinkert [1,3] ✉

Human deep space exploration is presented with multiple challenges, such as the reliable, efficient and sustainable operation of life support systems. The production and recycling of oxygen, carbon dioxide ($CO_2$) and fuels are hereby key, as a resource resupply will not be possible. Photoelectrochemical (PEC) devices are investigated for the light-assisted production of hydrogen and carbon-based fuels from $CO_2$ within the green energy transition on Earth. Their monolithic design and the sole reliance on solar energy makes them attractive for applications in space. Here, we establish the framework to evaluate PEC device performances on Moon and Mars. We present a refined Martian solar irradiance spectrum and establish the thermodynamic and realistic efficiency limits of solar-driven lunar water-splitting and Martian carbon dioxide reduction ($CO_2$R) devices. Finally, we discuss the technological viability of PEC devices in space by assessing the performance combined with solar concentrator devices and explore their fabrication via in-situ resource utilization.

Long-term space missions face similar challenges to the realisation of a green energy economy on Earth: solar energy systems are required to convert and store energy in the form of fuels, electricity and chemicals for day and night operation at high efficiency, stability and durability. At present, about 1.5 kW out of the 4.6 kW energy budget of the Environmental Control and Life Support System on the International Space Station (ISS) is consumed by the Oxygen Generator Assembly (OGA)[1], a photovoltaic (PV)-driven water electrolyser for electrochemical oxygen production. The high energy demand, resulting from the required electrochemical potential for the water oxidation reaction and the associated reaction overpotentials due to hindered gas bubble removal in the microgravity environment as well as the high total mass, makes the OGA unfeasible for application in future space architectures. Moreover, the OGA and the Carbon Dioxide Reduction Assembly currently in place on the ISS bear the challenge of being notoriously cumbersome and prone to breakdowns due to obsolete, inefficient, or aging compartments[2]. The lack of reliable and efficient life support hardware points to the need for new extra-terrestrial

oxygen and $CO_2$ recycling systems in order to realise space habitats on the Moon and Mars[3].

Contrary to PV-driven electrolyser systems, photoelectrochemical (PEC) devices integrate the processes of light absorption, charge separation and transfer as well as catalysis. Recently, it has been demonstrated that PEC devices—currently developed for sustainable solar-to-chemical energy conversion processes on Earth—can be utilised to produce hydrogen in microgravity environments at terrestrial device efficiencies[4]. PEC devices have also been demonstrated to be able to extend the temperature range of water-splitting to lower temperatures[5]. These advantages provide a motivation to investigate the application of PEC device architectures as well for oxygen ($O_2$) and carbon dioxide ($CO_2$) management in space, where the sustainable production and recycling of life-supporting chemicals will be essential for human survival. Given the stringent mass and volume constraints during space travel, they could initially be transported due to their compact, monolithic design, but ultimately also manufactured within the confinements of an extra-terrestrial settlement via In-Situ Resource

---

[1]Department of Chemistry, University of Warwick, Coventry CV4 7AL, UK. [2]Institute of Mechanical Engineering, Ecole Polytechnique Fédérale de Lausanne (EPFL), 1015 Lausanne, Switzerland. [3]ZARM – Center for Applied Space Technology and Microgravity, University of Bremen, 28359 Bremen, Germany. ✉e-mail: katharina.brinkert@warwick.ac.uk

Utilization (ISRU)[6,7]. The terrestrially researched PEC water-splitting and carbon dioxide reduction reactions ($CO_2RR$) bear furthermore the advantage that they can be tuned to produce hydrogen ($H_2$) and a variety of carbon-based fuels such as methane ($CH_4$), which can serve as liquid $CH_4$ in a rocket propulsion mixture ($LOx/LCH_4$)[8–10], or carbon monoxide (CO), which can be utilised, e.g., in the Fischer–Tropsch process for the synthesis of other hydrocarbon-based fuels and chemicals. Despite these advantages, the technological viability of PEC devices for space applications has not been assessed yet.

PEC water-splitting devices entail the use of integrated semiconductor-electrocatalyst systems which operate terrestrially at the so far highest reported long-term stability and efficiency[11–13]. These monolithic devices are characterised by the photocathode-driven hydrogen evolution reaction (HER) coupled with an anode-driven oxygen evolution reaction (OER)[4]. With the stipulated presence of ice water in the lunar Shackleton Crater and current efforts for establishing a settlement, we have chosen the Moon's surface as a location for a solar water-splitting device. Commonly reported literature device architectures for PEC water-splitting systems are H-cell configurations[14–18]. Likely scalable design options are however either membrane-embedded monolithic devices[19] or membrane-electrode assembly devices[20]. $CO_2R$ devices operate under a similar principle except that the cathode provides an array of $CO_2R$ reactions alongside the competing HER, dependent on the subject conditions and electrocatalyst of choice. State-of-the-art terrestrial $CO_2R$ devices utilise gas-diffusion electrodes (GDEs)[21–26] due to mass transport limitations associated with the low, aqueous diffusivity and solubility of carbon dioxide (33 mM under standard conditions of 298.15 K and 1 atm)[27]. Our rationale for a Martian-based $CO_2R$ device derives from the readily abundant supply of carbon dioxide in the Martian atmosphere (96%)[28]. The half-cell equations considered in this work include therefore the OER, HER and $CO_2Rs$ given below.

$$2H_2O \rightarrow O_2 + 4H^+ + 4e^- \text{ (OER)} \quad E^o = -1.229\,\text{V vs. RHE} \tag{1}$$

$$2H^+ + 2e^- \rightarrow H_2 \text{(HER)} \quad E^o = +0.00\,\text{V vs. RHE} \tag{2}$$

$$CO_2 + 2H^+ + 2e^- \rightarrow CO + H_2O \quad E^o = -0.11\,\text{V vs. RHE} \tag{3}$$

$$CO_2 + 8H^+ + 8e^- \rightarrow CH_4 + 2H_2O \quad E^o = +0.17\,\text{V vs. RHE} \tag{4}$$

Based on the device configurations used in the most efficient, terrestrial PEC device architectures[11], we utilise a multi-junction semiconductor photocathode coupled to an oxygen-evolving anode for our water-splitting device model. Our $CO_2R$ photo-absorbers are 'buried' PV cells, as there is no direct semiconductor-electrolyte interface present[29]. Figure 1a, b provides the energy diagrams of the two systems explored. Practically, it becomes necessary to increase the number of discrete in-series connected semiconductor photoabsorbers to surmount the realistically required electrochemical potential, which is a sum of the thermodynamic reaction potential and the required overpotentials. This results furthermore in a trade-off between the photoabsorber-junction limited current and the additional potential generated by multiple photoabsorbers being optically connected in series[30]. We model both, the high-performance realistic and theoretical thermodynamic limited scenarios for single-, tandem-, and triple-junction semiconductors for membrane-integrated monolithic and PV-driven GDE devices under terrestrial, lunar, and Martian solar irradiance cycles. Detailed thermodynamic and kinetic considerations are essential for models producing tangible device designs[31] as variations in the thermodynamic electrochemical potentials ($U_\theta$), catalytic exchange current density ($i_\theta$), charge transfer coefficient

($\alpha$), incoming photon conversion efficiency (IPCE), semiconductor series ($R_s$) and shunt ($R_{sh}$) resistances, the overall faradaic efficiency (FE), as well as the external radiative efficiency ($\eta^{ext}$) affect the overall device performance. Changes in these parameters cause significant realistic annual solar fuel yield divergences from ideal scenarios. A complete account of the parameters used in each model alongside model descriptions is presented in the Supplementary Sections I and II (Supplementary Tables 1 and 2, respectively). The operating environment for the simulated devices is—unless stated otherwise—assumed to follow standard conditions. The conditions are justified within the relevant modelling sections. This work seeks to establish the theoretical foundations for the application of PEC devices in habitats on the Moon and Mars and delivers the first foray into exploring the feasibility of utilising them for oxygen production and carbon dioxide recycling. The devices require a separate design for each target location, which can result in vastly different optimal configurations. This necessitates modelling the photon distribution on Moon and Mars, the simulation of a new series of standard Martian air mass (MAM) spectra as well as the determination of the irradiance and temperature cycles of the celestial surface[32]. These parameters are then fed into the solar-assisted electrochemical device designs. Moreover, we provide annual fuel and oxygen production rates and discuss the incorporation of solar concentrator technology to realise the technological and economic feasibility of solar-driven lunar water-splitting and Martian $CO_2R$. Finally, we provide an overview of the natural abundance of resources on the Moon and Mars for PEC device fabrication via ISRU.

## Results and discussion
### Atmospheric modelling
Radiative transfer modelling drives our characterisation of solar-driven electrochemical devices, simulating electromagnetic radiation propagating through different media[33]. The constituent molecules of the Martian atmosphere are responsible for the partial or complete attenuation of radiation and the solar flux that penetrates the atmosphere is subject to scattering by Martian dust and ice clouds. The trans-versing radiation will decrease in intensity if absorbed by matter, increase if matter emits energy, or alter direction when scattered[33]. The energetic distribution of photons incident at a given celestial surface utilised by semiconductors can be described through irradiance spectra between 280 and 4000 nm. We use the openly available library for radiative transfer (LibRadtran) to simulate the atmospheric radiative transfer, i.e., in our study, specifically the Martian solar irradiance spectrum as a function of different solar zenith angles[34] (Fig. 2a). We used LibRadtran's pseudo-spherical discrete ordinates radiative transfer solver[35] for the solution of the radiative transfer equation and the Mie routines[34] for the calculation of the atmospheric scattering properties. The required Martian vertical atmospheric density profile was compiled from data freely available at the Martian climate database (MCD)[36] and consists of a 0–250-km vertical profile range of temperature, density, pressure and concentration of $CO_2$, $O_2$, ozone ($O_3$), and $H_2O$ (Fig. 2b). The Martian dust particles were modelled with spectrally resolved refractive index data[37,38] and ice clouds[39]. It becomes evident that if, e.g., the terrestrial AM 1.5G (1000 $Wm^{-2}$) or the extra-terrestrial Martian MAM 0 (580 $Wm^{-2}$) spectra are used to calculate the efficiency of solar energy converting devices instead of the MAM 1.5 spectrum (369 $Wm^{-2}$), a substantial deviation of the device performance will be observed. Given the negligible difference in distances when considering the Sun-Earth and Sun-Moon length scale, we have taken the extra-terrestrial AM 0 spectrum as our standard lunar absorption spectrum. Of the limited available literature comparisons, two spectra did provide a significant correlation with our MAM results[40,41]. The

**a**

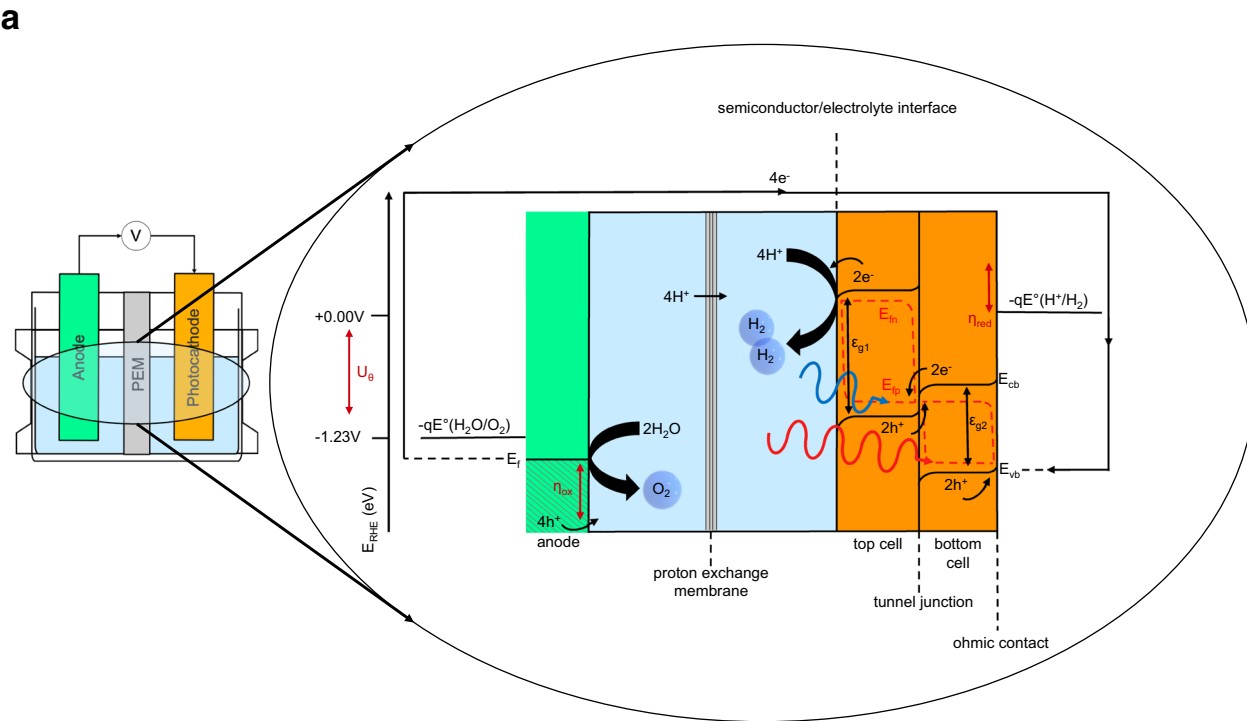

**b**

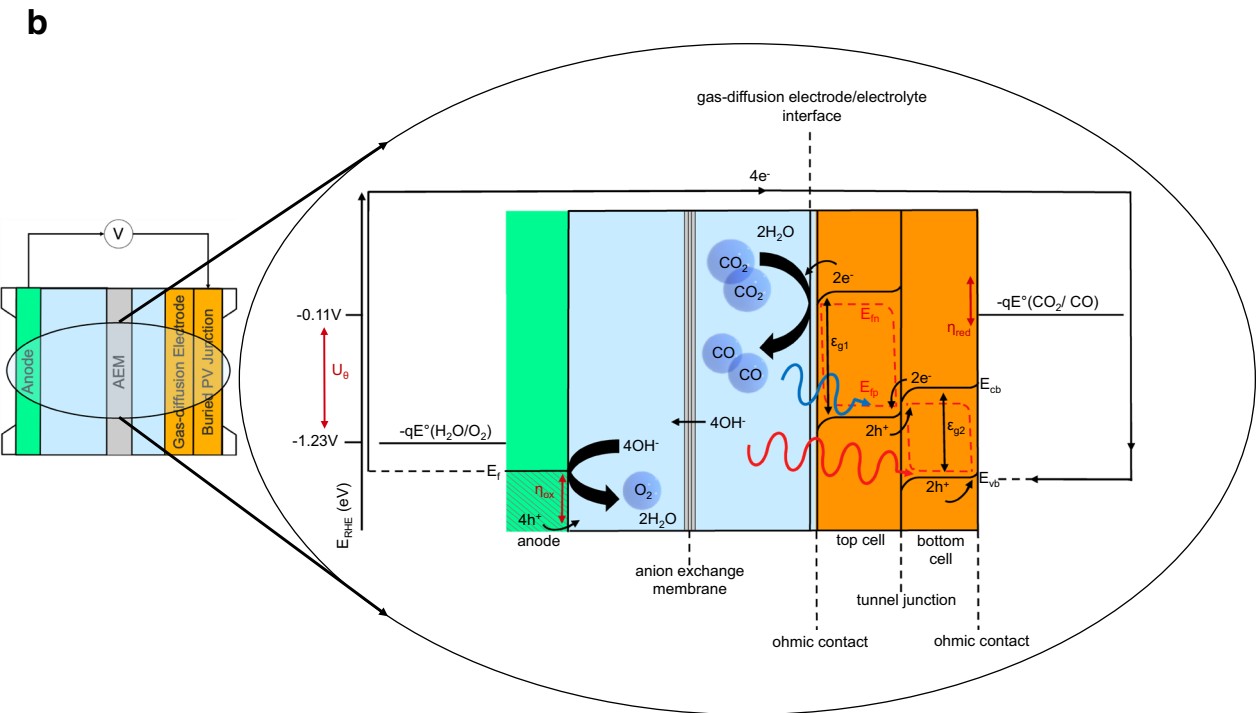

**Fig. 1 | Energy level diagrams. a** PEC (photoelectrochemical) energy level diagram, where the device consists of a metal anode and tandem p-n junction photocathode that forms an interface with the electrolyte. **b** GDE (gas-diffusion electrode)-based $CO_2$ device energy level diagram, where the device consists of a metal anode and buried p-n tandem semiconductor (SC) junction that forms an interface with the GDE. The conduction band ($E_{cb}$), valence band ($E_{vb}$), Fermi level ($E_f$), Quasi-Fermi electron level ($E_{fn}$), oxidative overpotential ($\eta_{ox}$), reductive overpotential ($\eta_{red}$), and thermodynamic electrochemical potential ($U_\theta$) are given versus the reversible hydrogen electrode (RHE).

differences between the intensity of the spectra can be attributed to the different solar zenith angle and locations used. Contrary to the other literature attempts at modelling the Martian absorption spectrum[40,41], we present ours as a range of solar zenith angles for appropriate comparison with lunar AM 0 and terrestrial AM 1.5G spectra. The American Society for Testing and Materials G-173 series reference solar spectra are used as AM 1.5G and AM 0 spectra, respectively[42].

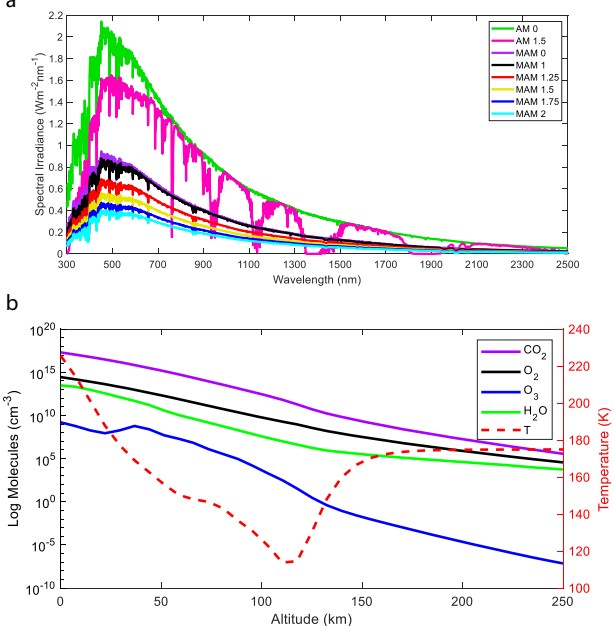

**Fig. 2 | Martian air mass (MAM) solar irradiance spectra. a** Simulated MAM spectra as a function of different solar zenith angles, where the solar zenith angle is the distance between a line perpendicular to the planetary surface and the position of the Sun in space. Each MAM spectrum is simulated with all input variables averaged over the course of a Martian year. The terrestrial AM 1.5G and AM 0 spectra are included for comparison[42]. **b** Yearly averaged, vertical molecular density profiles using the Martian climate database (MCD)[36] feeding into the radiative transfer calculations of the MAM spectra. The $y$-axis indicates the volumetric concentration (cm$^{-3}$) of molecules at a given altitude.

The Moon has a 1:1 orbital resonance with the Earth and this tidal locking results in a synodic period (lunar day) of 29.53 Earth days[43], where each lunar year entails 354.40 Earth days. This results in approximately 2 weeks of possible solar fuel and oxygen production followed by 2 weeks of darkness at the equator, increasing the requirement for reliable energy storage methods or the strategic positioning of solar-driven devices at the poles. A Martian year consists of 668.60 Martian days (sols), which each approximately equal to 1.03 Earth days[44]. These sols are much more characteristic to Earth's day and night (diurnal) cycle as they have similar day–night time ratios. The major deviation from Earth is that the Martian year lasts nearly twice as long, thus we can anticipate periods like the Martian winter to also persist almost twice as long as the winter on Earth. Consequently, there is a pressing requirement for any proposed Martian solar-energy conversion device to effectively sustain low solar irradiances, otherwise, there will be several months where these devices are simply unable to operate due to low incident solar radiation. Given the absence of a lunar atmosphere, temperatures on the Moon are prone to extreme and rapid fluctuations reaching from +120 to −233 °C at the shadowed lunar pole craters[45]. With the absence of reliable lunar–solar irradiance data, we assign a fitted, high-order Gaussian curve that is calibrated to result in 100% solar irradiance (1367 W m$^{-2}$) to the highest recorded lunar temperature. These irradiance cycles are given in Supplementary Fig. 1. Equatorial Martian temperatures have less extreme oscillations, but still fluctuate between −87 and −8 °C over the course of a Martian year[36].

## Solar water-splitting on the Moon

Our solar water-splitting model consists of a zero-dimensional equivalent circuit model, with electrochemical activation overpotential resistances described by Butler–Volmer electron transfer kinetics. Our photoabsorbers are optically connected in series, thus,

the overall photogenerated device current ($i_{ph}$) is given as the minimum current supplied by each junction, found through the integration of the characteristic solar flux ($\phi$). We quantify electrocatalytic activation overpotentials using the inverse-hyperbolic sign formulation of the Butler–Volmer equation[31]. The available voltage generated by the PEC device ($V_{PEC}$) is given as follows:

$$V_{PEC} = \left\{ \sum_n V_{PVi} - |\eta_{(cat,a)}| - |\eta_{(cat,c)}| - i_{PEC}R_{series} \right\} \geq U_\theta \quad (5)$$

Where $n$ represents the number of discrete junctions, $V_{PV_i}$ is the $i$ th junction photovoltage, $\eta_{(cat,a)}$ is the anode activation overpotential, $\eta_{(cat,c)}$ is the cathode activation overpotential, $i_{PEC}R_{series}$ the system series resistance, and ($U_\theta$) the thermodynamic electrochemical potential[31]. The reverse saturation current ($i_{\theta(PEC)}$) is then given by a detailed balance model[46] modified with the external radiative efficiency term ($\eta^{ext}$). Here, $h$ is the reduced Planck's constant, $q$ the elementary charge constant, $k_b$ is Boltzmann's constant, $T$ is the temperature, $c$ the vacuum speed of light, and $\varepsilon$ is the photon energy for a given wavelength. For multi-junction models, $\varepsilon_g(y)$ represents the larger bandgap of any junction layered above and $\varepsilon_g(x)$ represents the lower bandgap junction below:

$$i_{\theta(PEC)} = \prod_{i=1}^{n} \left( \frac{q}{4\pi^2 h^3 \cdot c^2 \cdot \eta^{ext}} \int_{\varepsilon_g(x)}^{\varepsilon_g(y)} \frac{\varepsilon^2}{e^{\left(\frac{\varepsilon}{k_bT}\right)} - 1} \cdot d\lambda \right) \quad (6)$$

The modified ideal diode equation[47] containing series ($R_s$) and shunt ($R_{sh}$) resistance terms results in a transcendental equation that is solved iteratively, with the diode ideality factor given as $z_D$. The device current ($i_{PEC}$) is given as:

$$i_{PEC} = i_{ph} - i_\theta e^{\left(\frac{q(V_{PEC} + i_{PEC}R_s)}{z_D k_b T}\right)} - \frac{V_{PEC} + i_{PEC}R_s}{R_{sh}} \quad (7)$$

The solar-to-chemical conversion efficiency ($\eta_{(STC)}$) is calculated using the integrated power density ($P_{in}$). The point of maximum efficiency occurs as the device voltage equals the overall required electrochemical potential ($U_{PEC}$), where $V_{op}\left(i_{op}\right) = U_{PEC}$[31].

$$\eta_{(STC)} = \frac{i_{op} \cdot (U_{\theta ox} - U_{\theta red}) \cdot FE(\%)}{P_{in}} \quad (8)$$

Here, $U_{\theta ox}$ and $U_{\theta red}$ are the oxidative and reductive half-cell potentials, respectively. Our PEC simulations to identify the thermodynamic limit present maximum possible ideal efficiencies as $IPCE$ and Faradaic efficiency ($FE$) values are taken to be 100%, $R_{series}$ is set to 0 Ω m$^2$, and both $R_{shunt}$ and $i_\theta$ values tend to infinity. This leads to negligible device resistances and Butler–Volmer catalytic overpotentials[31]. The thermodynamically limiting PEC efficiencies should be viewed as the limiting efficiencies attainable only by ideal devices and they essentially quantify how each semiconductor junction utilises the incident solar flux and can be used in thermodynamic 2nd law analyses.

Table 1 summarises the solar-to-hydrogen (STH) conversion efficiency and ideal bandgaps of each PEC model. Figure 3a summarises the diurnal operation of the realistic and thermodynamically limiting solar water-splitting models explored, where Pt and Ir-/Ru-oxide are used as cathodic and anodic catalysts, respectively. The 'turn on' thermodynamic model efficiency ramp is almost instantaneous when the ideal base electrochemical potential of 1.229 V is met. The slightly jagged lines after the maximum efficiency (Fig. 3b) can be attributed to fluctuations in solar irradiance and are caused by the high atmospheric attenuation within terrestrial models. When considering PEC devices based on the Moon, these contours are not as prevalent (Fig. 3c) because of the absence of tangible lunar atmospheric attenuation. The

**Table 1 | Summary of PEC model outputs**

| Model | Earth | Moon |
|---|---|---|
| Ideal case SJ | 30.7%, 1.59 eV | 28.4%, 1.59 eV |
| Realistic case SJ | 11.0%, 2.07 eV | 10.2%, 2.14 eV |
| Ideal case DJ | 40.3%, 0.51 eV, 1.39 eV | 38.1%, 0.57 eV, 1.32 eV |
| Realistic case DJ | 20.2%, 1.05 eV, 1.69 eV | 17.9%, 1.08 eV, 1.73 eV |
| Ideal case TJ | 28.1%, 0.31 eV, 1.04 eV, 1.68 eV | 29.4%, 0.31 eV, 0.90 eV, 1.58 eV |
| Realistic case TJ Earth | 15.9%, 0.79 eV, 1.27 eV, 1.84 eV | 14.6%, 0.83 eV, 1.26 eV, 1.88 eV |

Bandgap ($\varepsilon_g$) combinations are given alongside the solar-to-hydrogen conversion efficiency (STH, in %).
*SJ* singe-junction, *DJ* dual (tandem)-junction, *TJ* triple-junction.

addition of more photoabsorbers yields greater photovoltages to drive the electrochemical reactions, although, as especially evident in devices constituting three or more photoabsorbers (Fig. 3a), there is a decline in individual junction photogenerated current ($i_{ph}$) and therefore the overall efficiency as the solar flux accessible for each junction limits the overall device current ($i_{PEC}$). Fundamentally, the trade-off between $i_{PEC}$ and $V_{PEC}$ values are not significant for the tandem device as each photoabsorber is able to distribute the incoming solar flux effectively. The PEC results in Fig. 3 indicate that tandem semiconductor devices offer the best trade-off between high photocurrent density ($J_{H_2}$) and the necessary photovoltage to overcome realistic activation overpotentials.

Prevalent terrestrial atmospheric attenuation of high-energy photons (>3 eV) exists due to the presence of $H_2O$ and $CO_2$. Moreover, specific lower energy photon bands (<1.6 eV) are present because of ozone ($O_3$) and Rayleigh scattering (Supplementary Fig. 2). As such, the lunar spectrum (AM 0) has a higher relative integrated power density in the lowest and highest available photon energy regions in comparison to the AM 1.5G spectrum. This results in slightly lower STH efficiencies of lunar models than the ones obtained from equivalent bandgap terrestrial models, unless the lunar device has a junction that utilises the lower energy photon regions (<1.6 eV). For example, the ideal SJ device bandgap is 1.59 eV (Table 1), a higher proportion of AM 0 power density (45%) is distributed below this bandgap compared to AM 1.5G (43%) which contributes to the ~2% STH efficiency difference between the models. The difference in STH efficiency between terrestrial and lunar models decreases for the realistic SJ case as the ideal bandgaps increase. The AM 0 power distribution with photon energies below the ideal bandgap of 2.14 eV (68%) is closer to the distribution of AM 1.5G photons below the bandgap of 2.07 eV (67%). This trend is seen in all cases except for ideal triple-junction (TJ) devices where the minimum junction bandgap is 0.31 eV, corresponding to the upper limit of the wavelength range considered (4000 nm). Therefore, theoretically, all low-energy photons of each spectrum can be absorbed, and the lunar TJ device is predicted to be more efficient. The realistic TJ lunar case follows the previous trend of slightly lower STH conversion efficiencies relative to the terrestrial model, as the minimum bandgap increases significantly above 0.31 eV (to account for realistic overpotentials).

The yearly production yield was calculated by scaling the integrated power density of each spectrum (i.e. AM 0 for the Moon) with the integrated power density cycles given by Supplementary Fig. 1. The water-splitting scale-up capacity is 26 kg m$^{-2}$ of hydrogen produced per annum for realistic tandem-junction lunar devices (see Supplementary Fig. 3 for all model production yields). The introduction of a third or even more photoabsorbers is not only substantially less economically viable, but the annual production yields are also significantly lower than with tandem-junction devices, yielding 21 kg m$^{-2}$ of hydrogen produced per annum for a realistic triple-junction lunar device. The high-performance realistic values show the impact of sluggish

electron transfer kinetics that afflicts reactions which involve a larger number of electron transfer steps. This increase in half-cell electron transfer requirements translates to a significantly slower catalytic onset[31], evident in the OER which is shown in a gradual ramp to the point of maximum efficiency. Further model validations are discussed in Supplementary Section IV. As both, solar water-splitting and $CO_2R$ can require an OER electrocatalyst, improvements to the intrinsic quality of such electrocatalysts would yield substantial overall efficiency increases. Unassisted lunar–solar water-splitting is feasible due to the possible high achievable lunar photocurrents (20 mA cm$^{-2}$) and corresponding STH conversion efficiencies of 17.9%.

## $CO_2$ reduction on Mars

Unlike water-splitting devices, the most common $CO_2R$ devices are comprised of GDEs[22,25,48–54]. Supplementary Fig. 4 demonstrates the mass transport limited current density as a function of diffusion layer thickness. State-of-the-art GDE $CO_2R$ devices can offer current densities >200 mA cm$^{-2}$. We extend the previously developed one-dimensional analytical model[22] to include the light-assisted production of carbon monoxide and methane. The averaged channel length boundary layer thickness ($\delta^i$) of the linearised Poiseuille flow is given below with $D^i$ as the diffusion coefficient of species $i$ and $\nu$ as the average flow velocity along the direction of the channel[22]. $W_{channel}$ and $L_{channel}$ are the width and length of the flow channel, respectively.

$$\delta^i = 1.607 \frac{3}{4} \sqrt[3]{\frac{W_{channel} \cdot D^i \cdot L_{channel}}{\nu}} \quad (9)$$

Ion transport throughout the device is given by a simplified Nernst-Planck equation assuming averaged ion concentration, where $\epsilon$ is the catalyst porosity, and $a$ is the catalyst layer's volumetric surface area[22]. The governing conservation equation is then formulated as follows:

$$-D^0 \frac{\partial^2 [CO_2]}{\partial x^2} = \frac{a \cdot i_{\theta(CO_2R)}}{n_{e^-} F} \frac{[CO_2]}{[CO_2]_{ref}} \exp\left(\frac{-\eta_{CO_2R} \cdot F \cdot \alpha}{RT}\right) + \epsilon k^1_{\rightarrow} [CO_2] \cdot \langle [OH^-] \rangle \quad (10)$$

Here, $D^0$ is the Bruggeman corrected diffusivity, $\eta_{CO_2R}$ is the $CO_2RR$ activation overpotential, $F$ is Faraday's constant, $\alpha$ is the charge transfer coefficient, $R$ is the universal gas constant, $T$ is the temperature and $k^1_{\rightarrow}$ the forward rate constant of alkaline bicarbonate formation[22]. These equations are solved analytically, and the final current densities are given by the Butler–Volmer equation[22] with either Ag as a catalyst (for CO production) or Cu as a catalyst (for $CH_4$ production). The two different $CO_2R$ products are modelled by varying the electron transfer requirement ($n_{e^-}$) associated with each half-cell reaction. $i_{\theta(CO_2R)})$ is the catalytic exchange current density and $\alpha$ is the charge transfer coefficient associated with either Cu and Ag. Equations (5)–(8) are then utilised again to model the buried device PVs. The final current densities of HER and $CO_2R$ are given by:

$$i_{(H_2OR)} = i_{\theta(H_2OR)} \cdot \exp\left(\frac{-\eta^*_{H_2OR} \cdot F \cdot \alpha_{H_2OR}}{RT}\right) \quad (11)$$

$$i_{(CO_2R)} = \frac{[CO_2] \cdot i_{\theta(CO_2R)}}{[CO_2]_{ref}} \exp\left(\frac{-\eta^*_{CO_2R} \cdot F \cdot \alpha_{CO_2R}}{RT}\right) \quad (12)$$

The Nernst corrected potential is given by $\eta^*_{H_2OR}$ and $\eta^*_{CO_2R}$ for the reduction of $H_2O$ and $CO_2$, respectively. Parameter values for the kinetics, transport phenomena and cell dimensions are given in Supplementary Section II and Supplementary Table 2. Our analytical GDE-flow-cell model results are given in Fig. 4a. These results

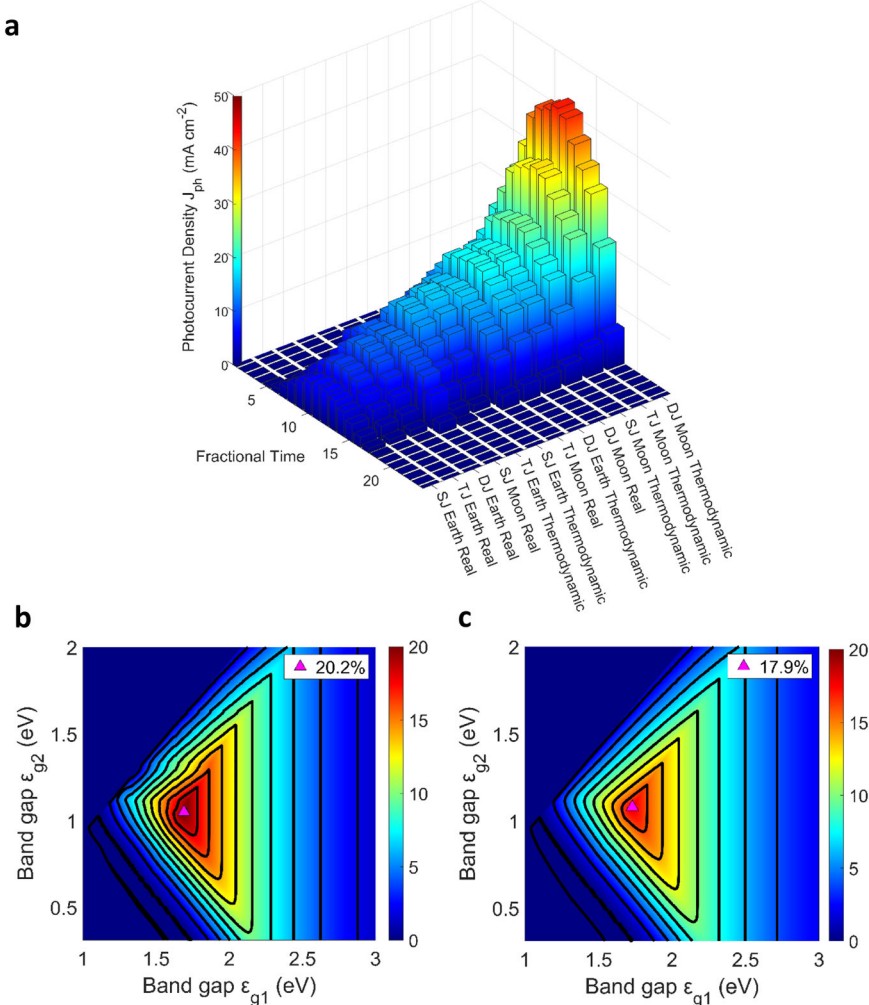

**Fig. 3 | Terrestrial and lunar–solar water-splitting models. a** Daily equatorial cycles of photocurrent density ($J_{H_2}$) of 12 water-splitting models covering single-junction (SJ), tandem (dual)-junction (DJ), and triple-junction (TJ) photoabsorber cathodes on Earth and Moon. **b** Realistic high-performance STH efficiency plots for terrestrial tandem-junction devices at 1000 W m⁻². **c** Realistic high-performance STH efficiency plots for lunar tandem-junction devices at 1367 W m⁻². Both tandem-junction device efficiency plots mark the highest solar-to-hydrogen (STH) efficiency configurations, and the contours indicate a change of Δ2% STH. The bandgap ($\varepsilon_g$) of the top (x-axis) and bottom (y-axis) photoabsorbers is varied to yield the STH for all viable semiconductor combinations. The fractional time for Earth corresponds to 1/24 of a full local day while for the Moon this equals to 1/24 of a lunar month.

initially bode well for Martian in-situ $CO_2R$ as the partial current density ($J_{CO_2R}$) values have the capability to reach ~200 and ~120 mA cm⁻² for carbon monoxide and methane, respectively (see Supplementary Figs. 5 and 6). Yet, when simulated under realistic terrestrial and Martian solar irradiance, it becomes evident that even the most effective $\varepsilon_g$ configurations struggle to surmount the extremely high cell potentials (>2.2 V) needed to drive these devices with solar irradiation. Thus, even though GDE devices could reach impressive partial current densities ($J_{CO_2R}$) under realistic solar irradiance cycles, these devices do not operate near their maximum potential (see Supplementary Fig. 3 for all model production yields). The difference in electron transfer requirements when producing carbon monoxide (2e⁻, Fig. 4b, c) or methane (8e⁻, Fig. 4d, e) significantly increases the activation overpotentials predicted by the Butler–Volmer equation. In turn, this results in much lower photocurrent densities for methane-producing devices, and subsequently, a further reduced fuel yield. Therefore, Martian space habitats may benefit from photoelectrochemically producing carbon monoxide and syngas in the first step which can then be used in a Fischer–Tropsch process for hydrocarbon synthesis. The lack of significant Martian atmospheric attenuation means that the power density distribution of the Martian spectrum is more closely correlated with the one of the AM 0 spectrum than with the AM 1.5G one (Supplementary Fig. 2). As the $CO_2R$ device models on average require higher open-circuit voltages ($V_{oc}$) than their water-splitting counterparts, there are fewer lower ideal bandgaps for $CO_2R$ available (Table 2). With the large bandgap requirement of single-junction (SJ) CO production, the Martian-based device is predicted to be more efficient than its terrestrial counterpart. With subsequent additions of medium bandgap junctions, this difference becomes skewed in favour of the terrestrial device (as the terrestrial device can utilise more of the mid-energy photon region where it has a higher relative power density).

## Environmental aspects—lunar and Martian dust

We also investigated the potential performance impact of lunar and Martian soil dust on light transmission as it is a general, significant threat to the performance of equipment and instruments. A 2D electromagnetic wave propagation model was developed in the frequency domain using the COMSOL wave optics module[55]. Maxwell's curl equations (Eqs. (13) and (14)) are combined in (15) to solve the electrical (**E**) and magnetic (**H**) field vectors for a discrete number of

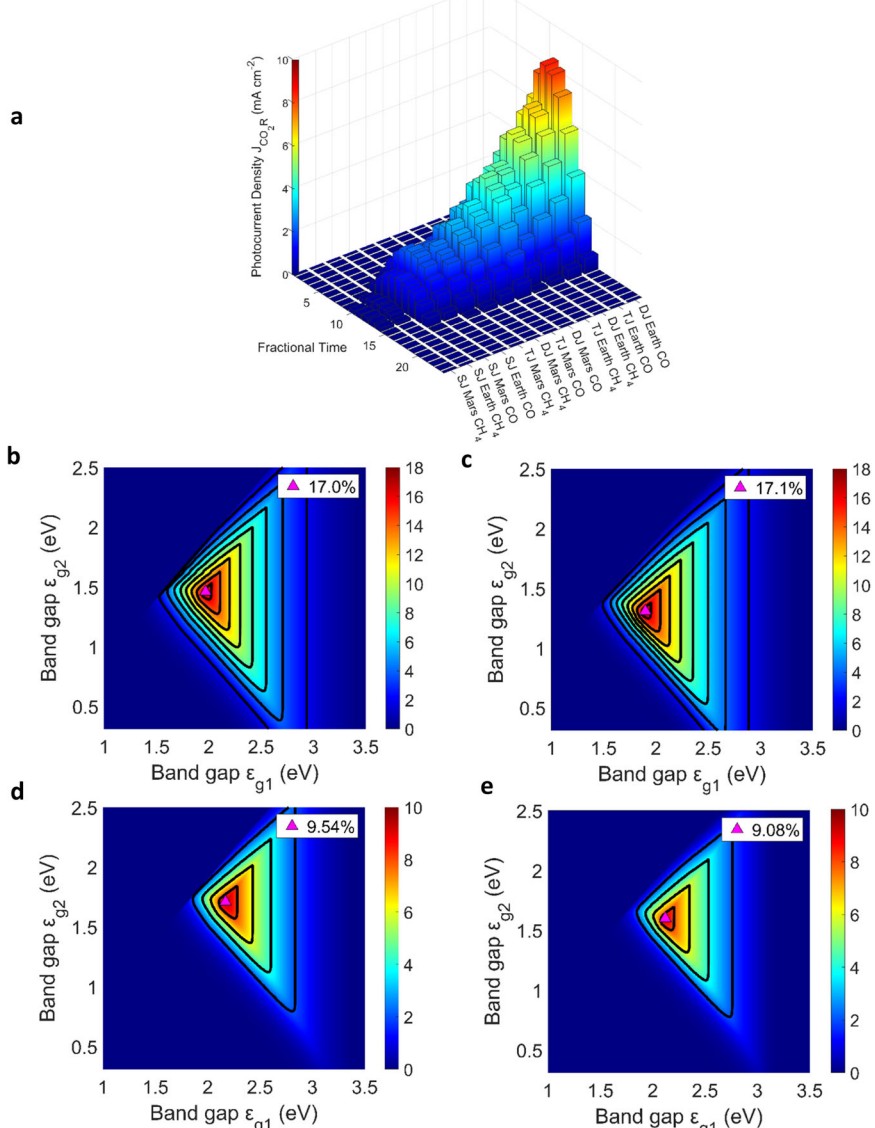

**Fig. 4 | Terrestrial and Martian CO₂R models. a** Daily equatorial cycles of partial photocurrent density ($J_{CO_2R}$) of all 12 CO₂R models covering single-junction (SJ), dual-junction (DJ), and triple-junction (TJ) photoabsorbers. **b** Realistic high-performance efficiency plots for terrestrial carbon monoxide tandem-junction devices at 1000 W m⁻². **c** Realistic high-performance efficiency plots for Martian-based carbon monoxide tandem-junction devices at 369 W m⁻². **d** Realistic high-performance efficiency plots for terrestrial methane tandem devices at 1000 W m⁻².

**e** Realistic high-performance efficiency plots for Martian-based methane tandem devices at 369 W m⁻². All tandem-junction device efficiency plots mark the highest solar-to-chemical (STC) conversion efficiency configurations, and the contours indicate a change of Δ2% STC conversion efficiency. The bandgap ($\varepsilon_g$) of the top ($x$-axis) and bottom ($y$-axis) photoabsorbers is varied to yield the STC for all viable semiconductor combinations. The fractional time for Earth and Mars corresponds to 1/24 of a full local day.

wavelengths using the finite element method[55].

$$\frac{\partial}{\partial t}\mathbf{B}_\lambda(r,t) = -\nabla \times \mathbf{E}_\lambda(r,t) \tag{13}$$

$$\frac{\partial}{\partial t}\mathbf{D}_\lambda(r,t) = \nabla \times \mathbf{H}_\lambda(r,t) - \mathbf{J}_\lambda(r,t) \tag{14}$$

$$\nabla \times \mu_r(\nabla \times \mathbf{E}_\lambda) - k_0^2\left(\varepsilon_r - \frac{j\sigma}{\omega\varepsilon_0}\right)\mathbf{E}_\lambda = 0 \tag{15}$$

Here, **B** is the magnetic flux density, **D** is the electric flux density, **J** is the electric current density, $r$ is position, $t$ is time, $\mu_r$ is the relative permeability, $\omega$ is the operating angular frequency, $c_0$ is the speed of light in vacuum, $\varepsilon_r$ is the relative permittivity, $\varepsilon_0$ is the permittivity of

free space $k_0$ is the wavenumber in vacuum, and $\sigma$ is the electrical conductivity. The COMSOL electromagnetic (EM) attenuation model was solved using the wave optics module. The model solves for the transverse electric (TE) wave through the electric field component in the $z$ direction out of the model $xy$-plane. The electric field component for the transverse magnetic (TM) wave is in the model $xy$-plane, and the magnetic field only has a component in the $z$ direction. The averaged TE and TM wave model results as a function of the dust deposition layer are given in Supplementary Fig. 1. The 2D model consists of an incident EM wave port permeating through a perfect vacuum and subsequently, through a variable-length dust layer flanked by floquet periodicity conditions on either side.

As the wavelength-dependent complex refractive index of lunar regolith is not available in the literature, we have undertaken a first approximation based on averaging the complex refractive index[56–58] of

**Table 2 | Summary of ideal bandgap ($\varepsilon_g$) combinations for PV-driven GDE devices**

| Model | $CO_2$ reduction (product: CO) | $CO_2$ reduction (product: $CH_4$) |
|---|---|---|
| Realistic SJ Earth | 4.44%, 2.64 eV | 0.98%, 3.04 eV |
| Realistic SJ Mars | 5.19%, 2.54 eV | 0.72%, 2.95 eV |
| Realistic DJ Earth | 17.0%, 1.46 eV, 1.97 eV | 9.54%, 1.71 eV, 2.17 eV |
| Realistic DJ Mars | 17.1%, 1.31 eV, 1.90 eV | 9.08%, 1.60 eV, 2.12 eV |
| Realistic TJ Earth | 13.7%, 1.26 eV, 1.62 eV, 2.11 eV | 7.56%, 1.55 eV, 1.85 eV, 2.29 eV |
| Realistic TJ Mars | 13.4%, 1.11 eV, 1.50 eV, 2.06 eV | 6.92%, 1.42 eV, 1.77 eV, 2.26 eV |

The bandgap ($\varepsilon_g$) combinations are given alongside the solar-to-chemical conversion efficiency (STC, in %).
SJ singe-junction, DJ dual-junction, TJ triple-junction.

the majority (> 96%) of regolith constituents according to the experimental analysis of several Apollo landing samples[59,60] (Supplementary Table 6). Reflectance ($\rho$) is related to the ratio of refractive indices ($n_i$) of two media:

$$\rho = \left[ \frac{n_{xi} - n_{yi}}{n_{xi} + n_{yi}} \right]^2 \qquad (16)$$

Given that the real component of the refractive index ($n_{yi}$) for a perfect vacuum is 1 and the average real component for lunar regolith components ($n_{xi}$) is ~1.6, the expected reflection is ~5%, which is consistent with the frequency domain model results of Supplementary Fig. 9. The main component affecting the performance of a PEC device is light absorption by dust deposition. The absorption coefficient ($\alpha$) is related to the imaginary refractive index component ($k$) by $4\pi k/\lambda$ and an increase in the imaginary component will have a substantial impact on the solar flux transmission through the respective dust layer. This difference distinguishes the results of the Martian and lunar regolith dust analyses (Fig. 5a, b) due to the larger absorption coefficient of lunar regolith. Figure 5b highlights the severe attenuation of high-energy photon transmission by Martian soil dust. A dust layer thickness of ~1 μm will result in an integrated power loss of roughly 50% for a semiconductor utilising the range of ~300–700 nm. Even greater current limitations will be found in devices constructed with greater than two semiconductor junctions, as the top junction of such devices tends to have a larger bandgap with each additional junction, which will suffer an even larger decrease in performance (a complete overview of the wavelength-dependent device impact is given in Supplementary Table 7). The devices that would result in the most stable operation—especially in dust storm periods—would be those with larger bandgaps. Interestingly, we already predict the need for larger bandgap tandem devices for operation on Mars for GDE device operation, owing to the inherently lower Martian solar flux. These devices would also be the most suitable design choice with respect to the continuous deposition of Martian dust (Fig. 5c).

Martian dust deposition will predominantly attenuate the higher energy photons between 300 and 600 nm (4.13 and 2.07 eV) until a dust layer thickness of roughly >1 μm is reached. This initial build-up will have minimal impact on devices consisting of fewer junctions, such as single and tandem, but will significantly limit the current density of devices using triple junctions with a larger top bandgap. This will also most likely render devices with quadruple and above semiconductor junctions practically inoperable without constant surface cleaning, compared to fewer junction devices. Our preliminary studies on the effect of lunar dust suggest that lunar regolith dust attenuation is more severe than Martian attenuation, a comparison between the individually weighted regolith dust components and Martian dust

complex refractive indices is given in Supplementary Fig. 8. A 100-nm lunar regolith layer will effectively render the lunar–solar flux to that of MAM 1.5 (Supplementary Fig. 9) and the surface cleaning requirement of PEC devices on the lunar surface is much more vital. Comparatively, a dust layer thickness of 500 nm will attenuate 99% of all incoming lunar–solar flux, while causing only 23% attenuation of incoming photons applicable to tandem-junction Martian devices (<2.12 eV). Due to the different ways dust is deposited on the lunar surface—with micrometeorites pulverising the lunar surface and solar wind imparting the resulting dust an electrostatic charge—a prediction of regular dust deposition is challenging and absent from the literature. The varying composition of lunar regolith[59] also causes a variance in device performance impact as it depends on the photoabsorber location and the different weighting of imaginary refractive index components. For instance, enstatite (pyroxene) positively correlates with iron content, which is evident in the iron abundant lunar mare regions[61], whereas bytownite and anorthite (plagioclase feldspar) inversely correlate with iron content as seen in the iron-deficient lunar highlands[61]. While the percentage of solar flux reflected by lunar minerals is going to be roughly similar (see the comparable real refractive index ($n$) of Supplementary Fig. 8), the absorption of solar flux can be expected to increase with the slightly higher imaginary refractive component of bytownite than enstatite.

This brief analysis shows that the development of, e.g., self-cleaning coatings is of uttermost importance not only for the application of PEC devices on the Moon and Mars, but essentially for all solar harvesting technologies.

## Engineering outlook

One technology that could be incorporated with solar-assisted oxygen and fuel production devices on the Moon and Mars is solar concentrators, enabling larger production rates and higher power density devices[62]. Solar concentrations of up to 100× (the solar concentration factor is defined as the ratio of radiant power density at the reactor aperture versus the radiant power density arriving from the Sun) can be generated with single-axis tracking concentrator technologies, while concentrations up to and above 1000× can be achieved with dual axis tracking technologies. Recent literature demonstrates the use of concentrated solar light with irradiation fluxes up to 450 kW m$^{-2}$ (450 times the integrated power of the AM 1.5G spectrum) terrestrially, which in turn induced a partial current density for the conversion of $CO_2$ to carbon monoxide of 175 mA/cm$^2$[,63]. Concentrated PEC research has to tackle a variety of challenges such as non-linear increases in activation overpotential, thermal management issues (photoabsorber recombination losses, membrane dryout), and accelerated material degradation[62], but they represent a viable option for compact, high power density solar devices in space.

To improve the practicality of Martian-based PEC devices, we designed a solar concentrator that would enable each tandem-junction device to reach a partial photocurrent density benchmark of 100 mA cm$^{-2}$. In order to achieve a partial photocurrent density of 100 mA cm$^{-2}$, the devices must generate an increased open-circuit voltage of 2.64 and 3.05 V for carbon monoxide and methane production, respectively. Thus, the optimal bandgap configurations for both Martian-based $CO_2R$ devices increase. The ideal bandgaps for Martian carbon monoxide production equal now 1.57 and 2.10 eV (previously 1.31 and 1.90 eV). For Martian methane production, the ideal bandgaps straddle 1.81 and 2.31 eV (previously 1.60 and 2.12 eV). It is important to note that the bandgap optimisation has to occur for each PEC device coupled to a solar concentrator individually. If the solar concentrators become inoperable at some point, the higher optimal bandgaps will produce negligible current densities under 1 sun Martian irradiance. Moreover, inherent to the use of concentrated solar irradiance is also a significantly higher impact of series resistance. The fill factor for terrestrial tandem CO production under 1 sun (79%

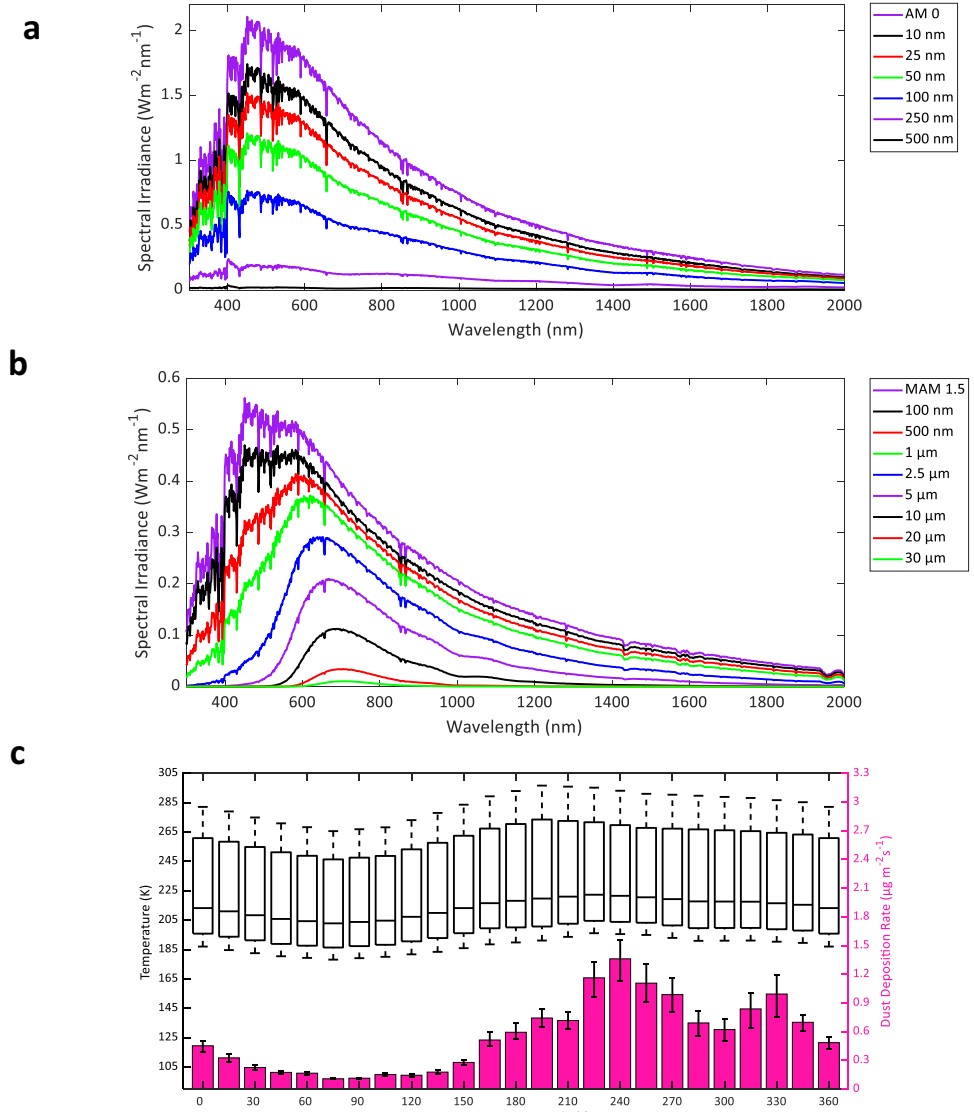

**Fig. 5 | Spectral irradiance impact of lunar and Martian regolith dust.**
**a** COMSOL electromagnetic wave propagation simulations multiplied by the AM 0 spectrum[42] to yield the effective lunar surface solar spectrum with specific regolith dust layer thicknesses. **b** COMSOL electromagnetic wave propagation simulations multiplied by the simulated MAM 1.5 spectrum to yield the effective Martian surface solar spectrum with specific Martian dust layer thicknesses. **c** Yearly temperature and dust deposition rates are shown as a function of areocentric longitude on Mars. Data are used from the Martian Climate Database (MCD)[36]. The dust deposition error bars indicate the variance at a given areocentric longitude. The temperature is displayed as a box plot with the median average indicated.

with 15 mΩ m$^{-2}$) was significantly reduced without altering the series resistance when under 25 suns (24%). We therefore reduced our series resistance term until the fill factor was no longer a severe bottleneck for the device efficiency (79% with 0.1 mΩ m$^{-2}$). Knowing the solar concentrator factor needed to induce 100 mA cm$^{-2}$ in each device, $C_{SC}$, we can calculate the area of a solar concentrator, $A_{SC}$, needed to achieve these partial current densities. The optical efficiency of the solar concentrator was taken as 80%[64]. Tabulated values are given below (Table 3), where $DNI$ is the incident direct normal irradiance. It becomes evident that by using concentrated light (at concentration factors between 5 and 35), we can vastly improve the partial photocurrent density of our poorest performing realistic tandem-junction device for Martian methane production from 3.10 to 100 mA cm$^{-2}$ with a solar concentrator area of 43 m$^2$. It is however important to note that only the photocurrent density of the device increases; a small $V_{oc}$ will still lead to a low overall device power output.

The natural abundance of materials which can be used as electrocatalysts or semiconductors varies quite significantly between the Earth, Moon, and Mars. Elements deemed too scarce for the terrestrial industry or materials which are inefficient under AM 1.5G solar irradiance may in fact be well suited to be utilised on different planets or moons for the construction of PEC devices. Supplementary Table 8 summarises the natural abundance of photoelectrochemically relevant

**Table 3 | Solar concentrator parameters**

| Model | $DNI$ (W m$^{-2}$) | $A_{SC}$ (m$^2$) | $C_{SC}$ | $R_{SC}$ (kg m$^{-2}$) |
|---|---|---|---|---|
| Lunar–solar water-splitting | 1367 | 7 | 5.4 | 91 |
| Martian CO production | 369 | 33 | 26 | 1491 |
| Martian CH$_4$ production | 369 | 43 | 34.5 | 859 |

Parameters used to design solar concentrators given an optical efficiency of 80%[64], calculated as the fraction of radiant energy that is incident on the photoelectrode surface relative to the magnitude of radiant energy that reaches the solar concentrator. Parameters are given for each device in order to produce a partial current density of 100 mA cm$^{-2}$. The chosen devices consist of optimised tandem-junction semiconductors. The annual rate of fuel production ($R_{SC}$) is given assuming the target of 100 mA cm$^{-2}$ is met for 1/3 of the year.

**Table 4 | Engineering guidelines for realistic solar-assisted oxygen and fuel production devices**

| Model | Model $\varepsilon_g$ | Semiconductors | Electrocatalysts OER, $H_2$/$CO_2$R |
|---|---|---|---|
| DJ Moon ($H_2$ production) | 1.08 eV, 1.73 eV | Si (1.10 eV), $Al_{0.25}Ga_{0.75}As$ (1.73 eV) | $IrO_2$, Pt |
| DJ Mars (CO production) | 1.57 eV, 2.10 eV | CdTe (1.50 eV), $Al_{0.90}Ga_{0.10}As$ (2.11 eV) | $IrO_2$, Ag |
| DJ Mars ($CH_4$ production) | 1.81 eV, 2.31 eV | $Al_{0.30}Ga_{0.70}As$ (1.80 eV), GaP (2.26 eV) | $IrO_2$, Cu |

Model bandgaps ($\varepsilon_g$) are those obtained via realistic device models combined with solar concentrators. The semiconductor column specifies the closest abundant material available on Moon or Mars that can be used for the design with the respective, intrinsic $\varepsilon_g$. One can use this table in conjunction with Supplementary Table 8 (natural resource abundances for the Earth, Moon, and Mars) to generate guidelines for an ISRU PEC design. The electrolytes correspond to 1 M $H_2SO_4$(aq) and 1 M $KHCO_3$(aq) for water-splitting and $CO_2$R, respectively.
*DJ* dual-junction.

elements as composites of semiconductors and electrocatalysts on Earth, Moon and Mars. It becomes evident that the in-situ utilisation of elements on both, the Moon and Mars, is feasible for the construction of PEC devices. Particularly interesting is the possibility of designing devices with terrestrially precious, but highly efficient electrocatalyst materials such as Pt and Rh, which allows the approach of thermo-dynamically limiting device efficiencies. Table 4 summarises the optimal bandgaps of PEC devices for solar water-splitting and $CO_2$R on Moon and Mars with suggestions for suitable electrocatalysts and semiconductor materials based on the lunar and Martian availability (Supplementary Tables 8 and 9).

Current competing space technology for oxygen production includes among others the Mars Oxygen In-Situ Resource Utilisation Experiment (MOXIE)[65], the Micro-Ecological Life Support System Alternative (MELiSSA)[66] and lunar regolith electrolysis[67]. Despite providing very interesting, promising concepts, they remain scientifically and technically challenging. MOXIE utilises the solid oxide electrolysis of $CO_2$ which requires extremely high reaction temperatures (>800 °C)[65]. Besides the high-energy input, MOXIE requires an atmospheric compression to 0.7 bar[65], which—if not maintained—can result in coking (carbon deposition) that subsequently causes cathode instability and fracturing[68]. MOXIE currently operates at ≈0.5% of the scale needed to produce sufficient enough oxygen for a four-person trip to Mars[65]. Moreover, the current MOXIE hardware requires revision if it be scaled up for reliable use on Mars[68]. Extraction of oxygen from lunar regolith can proceed through three main routes: (i) vacuum-pyrolysis methods operating at very high temperatures (1000 °C)[62], (ii) reactive gas-based methods[67] that require a continuous supply of gaseous reactants not readily available in deep space such as $H_2$, $F_2$, or $CH_4$ or (iii) an electrolysis-based extraction that still requires relatively high input energy to maintain reaction temperatures (100 °C) as well as exotic electrolytes like 1-ethyl-3-methylimidazolium hydrogen sulfate ([ENIM][$HSO_4$])[67]. Experimental lunar regolith extraction also suffers from a scarcity of real lunar samples in regular terrestrial trials[67]. Our work on approximating the refractive index of regolith has also highlighted the local variance in regolith composition which furthermore complicates process standardisations. MELiSSA is a closed life-support system that can produce food, water, and oxygen[66]. A closed-loop life-support is as good as the weakest link[66], signifying that the system needs to be very efficient and well-maintained throughout. Challenges for the implementation of PEC devices on the Moon and Mars remain without doubts such as the required long-term stability, a high-energy efficiency and production rate. A more detailed technoeconomic analysis is furthermore required to complement feasibility investigations and potential weight advantages as previous analyses lack technical details[69]. Oxygen production via unassisted PEC systems can however be carried out at room temperature in aqueous electrolytes that are suitable to be housed in temperature-controlled space habitats[70]. The device construction can draw from a variety of semiconductors and electrocatalyst materials that are available on the Moon and Mars (Supplementary Tables 8 and 9) and the required materials can eventually be produced via ISRU. Moreover, we have previously demonstrated that PEC devices can work

efficiently in microgravity[4,71] and our theoretical analysis suggests that it can suitably be scaled up.

## Viability of photoelectrochemical devices in space

We have demonstrated the thermodynamically limiting and realistic high-performance scenarios for unassisted H-cell type PEC water-splitting under terrestrial and lunar–solar irradiance as well as GDE-flow-cell $CO_2$R under terrestrial and Martian solar irradiance. Within our modelling framework, we have shown that tandem-junction photoabsorber cells are the most effective configurations for terrestrial, lunar, and Martian environments when examining realistic long-term solar-to-chemical conversion efficiencies. Overall, lunar PEC water-splitting (tandem devices >20 mA cm$^{-2}$) possesses a very high capability for hydrogen and oxygen production, whereas Martian-based PEC $CO_2$R (tandem devices <9 mA cm$^{-2}$) requires the coupling to solar concentrators to overcome the inherently low solar irradiance on Mars and to become technically viable for oxygen production and $CO_2$-to-fuel conversion. Solar-driven GDE devices are in these conditions not able to reach their full potential. Further electrochemical $CO_2$R research—for terrestrial and space applications—should be directed towards in-depth solar concentrators modelling to raise the photoelectrode/PV output or revisit conventional H-type $CO_2$R devices which could be more effective for $CO_2$R at low current densities. Nevertheless, from both, experimental and theoretical perspectives, challenges and questions regarding the application of PEC devices on the Moon and Mars remain. Further experimental and theoretical studies could investigate (i) the capacity for exploiting low ambient Martian temperatures to examine the performance of H-cell designs that can overcome the solubility-limiting mass transport, (ii) the feasibility of incorporating solar concentrators to increase the overall solar-chemical production rates, and (iii) a technoeconomic comparison between low-temperature H-cell devices for $CO_2$R versus GDE devices with the incorporation of solar concentrator technology. In addition, our analysis presented here does not consider all environmental challenges a device faces in space, such as space radiation, extreme temperature fluctuations or reduced gravitation. Although high long-term efficiencies and power densities of PEC devices are still integral parts of ongoing terrestrial research efforts, we have shown that the application of these devices could go beyond Earth and potentially contribute to the realisation of human space exploration. Moreover, it opens the possibility of exploring (photo-)electrochemical devices as well in other harsh environments such as the terrestrial polar regions[5] as electrochemical devices have previously been demonstrated to work at low temperatures, where they offer potential benefits to steering product selectivity and overcoming gas solubility limits[72–74].

## Data availability

All data generated in this study are provided in the manuscript or the Supplementary Information file.

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

## Acknowledgements

B.R., K.B. and S.H. would like to thank the European Space Agency and Warwick's Analytical Science CDT for the generous support of B.R.'s PhD studentship. K.B. would like to thank the German Aerospace Center (DLR) with funds provided by the Federal Ministry for Economic Affairs and Energy (BMWi), Germany, under Grant No. DLR 50WM2150 (project *LiMo*) for the generous support. The activity was selected via the *Open Space Innovation Platform* (https://ideas.esa.int) and carried out under a programme of and funded by the European Space Agency (Idea: I-2021-02002). All authors would like to thank Dr Douglas Rickman and Dr Rostislav Kovtun (NASA), Dr Michael Wolff (Space Science Institute), Ömer Akay (ZARM), Dr Christel Paille and Dr Brigitte Lamaze (ESA) for helpful personal correspondence.

## Author contributions

B.R. carried out the theoretical framework developed and simulations, supervised by S.H. and K.B. B.R. wrote the article and with support from S.H. and K.B., K.B. and S.H. finalised it. S.H. and K.B. conceptualised the project.

## Funding

## Competing interests

The authors declare no competing interests.
