## [Peer Review File · Nature Communications]

REVIEWER COMMENTS

Reviewer #1 (Remarks to the Author):

This paper presents a theoretical study on the potential of photoelectrochemical (PEC) devices for O₂ production on Moon via water splitting and CO (as a fuel) production on Mars via CO₂ reduction. The authors computed solar irradiance for Moon and Mars (in comparison with that for Earth) and then applied existing models (ref [27] for water splitting; ref [18] for CO₂ reduction) to complete the energy efficiency calculations for both cases. While I found the work interesting, I do have a number of concerns and questions pertaining to the nature of the contribution in the context of space applications (which appears to be the focus of this work) and some technical details.

I don't have specialist knowledge about space applications but wondered about the significance of the modelled technical schemes in this context. I believe the following issues need to be addressed convincingly so that the importance of this work can be established.

(1) For the supply of O₂, which is clearly important, to what extent is the photoelectrochemical (PEC) route more preferable to the combination of separate photovoltaic and electrolysis (the latter seems to be what's already used in space stations), considering both efficiency and reliability, and the broader need for power supply in say a space station (which could justify a centralised PV to support multiple power uses)?

(2) For CO₂ reduction, it's generally considered as a process worth exploring for the circularity of CO₂ in the energy system of the world, motivated by the need for reducing greenhouse gas emissions; this is perhaps not very relevant for applications on Mars and it is unclear why the production of a carbon-based fuel would ever become a priority in this context.

(3) As a relatively minor aspect of the work, the authors presented the resource abundance on Moon and Mars for the minerals involved in building the PEC devices, which showed superior availability of several materials compared to those on Earth. I wonder what's the intention of this comparison – does it imply that the manufacturing of the PEC devices could use materials obtained on Moon or Mars within the timeframe of relevance? Would this be necessary (if only say a dozen of PEC devices are needed, so supply on Earth would not be a constraint) or realistic (if instead a high demand is to be met, which would need significant mining/transport)?

Some questions on technical/presentation details:

(4) Within the context of space applications on Moon and Mars, it is not entirely clear what operating conditions have been assumed for the water splitting device and the CO₂ reduction device, as adopted in the mathematical modelling, in terms of temperature, pressure, CO₂

concentration, and why. SI Table 2 gave some information, but it's not complete and also without justification.

(5) Page 2, 4th line above equation (1), 33mM was mentioned as the solubility of CO₂. I believe this is true specifically for 25°C and pCO₂=1bar, which is not necessarily applicable for a GDE.

(6) Page 6, equation (5) and those introduced thereafter, not all symbols have been explained.

(7) Page 7, Equation (6): what's the relationship between V_{PEC} there and V_{op} (and between i_{PEC} and i_{op}) in SI equation (18)? How did the activation overpotentials influence the efficiency calculation (was it through the calculation of i_{PEC})?

(8) Page 7, Equation (7), about the notation: ρ_{in} or rather p_{in} for integrated power density?

(9) For the CO₂ reduction part, what was the assumed feed CO₂ purity? 100%? How was the GDL component modelled? Were the rate of gas-liquid mass transfer and the rates of aqueous phase reactions modelled, or instead these processes were all treated by phase or chemical reaction equilibrium?

Reviewer #2 (Remarks to the Author):

Ross et al investigate theoretical conversion efficiencies and device concepts for photoelectrochemical (PEC) water splitting and CO₂ reduction under the conditions found on Moon and Mars. For this, they model solar spectra and PEC devices under the conditions found there, finally deriving potential yearly production yields and engineering considerations. The topic of their work is certainly a very interesting subject and therefore in principle suitable for publication in Nature Communications. There are, however, a number of points regarding presentation, discussion and literature context, that should be addressed:

1. The statement on page 2, 'PEC water-splitting devices [...] so far highest reported long-term stability and efficiency.' is a bit unclear, as the highest efficiencies mentioned in the references are typically not highly stable, at least when compared to PV-electrolysis. This should be put in an appropriate context for the reader without a PEC water splitting background.

2. Figure 1 is somewhat inconsistent, as it shows an illumination source and charge-carriers, implying a device under illumination. Yet the Fermi level is shown for a device in the dark. Furthermore, the arrow for the U_{θ} (thermodynamic electrochemical potentials) looks a little bit like it is aligned with the band edges, yet this should be Quasi-Fermi-level splitting.

3. It would be illustrative to also show the AM 1.5G (and maybe also AM 0) spectrum in Fig 2a for comparison, as the Authors also refer to differences in those spectra later on.

4. On page 6, the Authors mention the low temperatures on Moon and Mars. But then, they do not seem to include temperatures in their model? This might be a very important point distinguishing PEC approaches from photovoltaic(PV)-electrolysis approaches, as PEC allows to extend the temperature range of the reaction into lower temperatures when compared to a PV system with separate electrolyser (see e.g. [Kolbach et al, Energy Environ. Sci. 14, 2021]).

5. Page 7, paragraph 'We utilize...'. This needs to be made more clear: Explain the rather uncommon definition of 'partial operating current'. Is this the intersection of the current-voltage curve of the solar cell with the catalysts' operating current? Which temperature is assumed in the model? What is meant by 'accounting for both, ohmic and electrocatalytic overpotentials'? In the 'ideal' cases later on, neither catalysis nor ohmic losses seem to be taken into account. This should be made more clear, as the 'realistic case' still seems to be quite idealised with very good catalysts and absorbers. So please explain the different levels of idealisation more clearly.

6. There should be some discussion to what extent and by which materials the assumed solar cell characteristics can be achieved in experiment.

7. 'The bandgaps of all junctions must sum together with a greater open-circuit voltage...'. This is a strange statement, as bandgaps come in energy (eV), voltage in the unit (V).

8. The paragraph 'The water splitting scale...' on page 8 is unclear and should be rewritten. Is the yearly production yield something calculated with daily cycles such as in [DOI:10.1039/D2SE00561A or DOI:10.1039/D0SE01207F] or is it simply taken by using the yearly average from Fig. 2 and scaling the average spectra over the course of a 'standard day'? 'High photocurrents (>20 mA/cm²)' and STH of 17.9% is contradictory, as 20 mA/cm² is more than 20% STH. The catalysts for 'realistic exchange current density' are the ones mentioned in the SI? Please elaborate a bit more here in the main manuscript.

9. The Authors go into details of an "Engineering outlook", but for a space mission, the carry-along-mass of the system is certainly also an important aspect, which should be discussed. See for instance a similar article on PV-powered chemical conversion on Mars [Abel et al, Front. Astron. Space Sci. 9, 2022]. I would guess a PEC system will be heavier than a PV-electrolysis system?

10. A potentially noteworthy literature context might be that natural, dual-junction PEC water splitting might have taken place on Mars: DOI:10.2138/am.2012.4211.

Reviewer #3 (Remarks to the Author):

- What are the noteworthy results?

The manuscript presents some noteworthy results on how one could produce oxygen on Mars by splitting CO₂ and on the Moon by splitting H₂O. The idea is not new, but the analysis is.

- Will the work be of significance to the field and related fields?

Yes, marginally. The comparison of the performance to that on earth is interesting. However, it is not evident that potential temperature effects were realistically evaluated.

- How does it compare to the established literature? If the work is not original, please provide relevant references.

As oxygen production on Mars has been demonstrated using high-temperature electrolysis before this work could be published it would be important to first acknowledge that work and second show why the approach here is preferable to what has already been demonstrated. That technology is not even mentioned.

- Does the work support the conclusions and claims, or is additional evidence needed?

The manuscript seems to recognize the dust in the Martian atmosphere in the atmospheric modeling but not the implication of how to keep dust from settling on the device and degrading its performance. Neither does the manuscript recognize what would happen in a Martian dust storm, which could eliminate production for weeks.

- Are there any flaws in the data analysis, interpretation, and conclusions? Do these prohibit its publication or require revision?

Addressing a comparison to state-of-the-art technology does require revision. Addressing dust and dust storms on the performance, not just the atmospheric modeling must also be addressed before considering publication.

- Is the methodology sound? Does the work meet the expected standards in your field?

YES

- Is there enough detail provided in the methods for the work to be reproduced?

YES

Department of Chemistry
University of Warwick
Gibbet Hill Road
Coventry CV4 7AL

Response to reviewer comments

Reviewer #1

“This paper presents a theoretical study on the potential of photoelectrochemical (PEC) devices for O₂ production on Moon via water splitting and CO (as a fuel) production on Mars via CO₂ reduction. The authors computed solar irradiance for Moon and Mars (in comparison with that for Earth) and then applied existing models (ref [27] for water splitting; ref [18] for CO₂ reduction) to complete the energy efficiency calculations for both cases. While I found the work interesting, I do have a number of concerns and questions pertaining to the nature of the contribution in the context of space applications (which appears to be the focus of this work) and some technical details.

I don't have specialist knowledge about space applications but wondered about the significance of the modelled technical schemes in this context. I believe the following issues need to be addressed convincingly so that the importance of this work can be established.

- 1. For the supply of O₂, which is clearly important, to what extent is the photoelectrochemical (PEC) route more preferable to the combination of separate photovoltaic and electrolysis (the latter seems to be what's already used in space stations), considering both efficiency and reliability, and the broader need for power supply in say a space station (which could justify a centralised PV to support multiple power uses)?”*

This is an interesting comment that will require further studies beyond the scope of this manuscript. A few initial points for consideration are listed below.

Existing literature of technoeconomic analyses of PEC and PV-electrolyser devices focus on terrestrial \$/kg_{H2}. This weighting of factors such as cost is of smaller concern to space agencies than designing lightweight, compact devices which operate at high energy efficiency and provide long-term stability. PEC systems occupy a smaller volume, operate at room temperature in aqueous solutions and can readily be incorporated in temperature-controlled Moon and Martian habitats [ESA Moon Village CDF Study Report: CDF-202(A), 2020]. Thermally-coupled PEC devices have also been shown as being more efficient than PV-electrolysis devices at lower temperatures, paving the way for low-temperature applications [Kolbach *et al*, *Energy Environ. Sci.* 14, 2021]. Moreover, proposed oxygen extraction from lunar regolith requires temperatures around 900°C [Ö. Akay *et al*, *npj Microgravity* 56, 2022] which requires significantly more input energy than a PEC route. A sufficient performance analysis and prediction of PV-electrolyser systems is however absent in space environment as the analyses is complicated by multiple parameters and their impact which are difficult to study independently from each other in space e.g., reduced gravitation and space radiation. In order to fully compare the two approaches e.g., for oxygen production, theoretical models are required which take into consideration the impact of reduced gravitation on the specific

Department of Chemistry
University of Warwick
Gibbet Hill Road
Coventry CV4 7AL

overpotentials, mass transfer processes and technical aspects such as weight and volume of the devices. We are currently working on such a comparison by extending our existing PEC models to PV-electrolyser systems. It requires however a full model of the electrochemical interface processes including gas bubble formation which need to be integrated with the models developed as part of this work.

We have included a state-of-the-art comparison of O₂ production pathways in our 'Engineering Outlook', briefly addressing other competing O₂ production technology such as MOXIE and MELISSA and the fact that the current Oxygen Generation Assembly (OGA) on the International Space Station (ISS) will not be able to be used for oxygen production on long-term space missions, motivating research into new technologies. [e.g., <https://ntrs.nasa.gov/citations/20160014553>].

2. *"For CO₂ reduction, it's generally considered as a process worth exploring for the circularity of CO₂ in the energy system of the world, motivated by the need for reducing greenhouse gas emissions; this is perhaps not very relevant for applications on Mars and it is unclear why the production of a carbon-based fuel would ever become a priority in this context."*

We appreciate the comment. Key challenges deep space exploration face are related to mass and volume restrictions during transport. If a return trip to Mars should be made possible, in-situ rocket fuel production will be necessary as resupplies cannot be carried on missions to Mars. Also, vehicles on Moon and Mars require fuel. NASA and ESA are both pursuing in-situ resource utilisation initiatives [https://kiss.caltech.edu/final_reports/ISRU_final_report.pdf], and actively research and develop LOx/methane based rocket propulsion systems [<https://ntrs.nasa.gov/citations/20170005557>]. These propulsion systems have been designated as a leading candidate for future deep space missions, as liquid methane has a six times higher density than hydrogen which makes it a more favourable candidate in deep space missions than hydrogen-based propulsion systems. Therefore, given the high Martian atmospheric abundance of CO₂ and the need to produce LOx/methane in-situ on Mars, the reduction of CO₂ to hydrocarbon fuels is a potential solution to make a return trip possible.

3. *"As a relatively minor aspect of the work, the authors presented the resource abundance on Moon and Mars for the minerals involved in building the PEC devices, which showed superior availability of several materials compared to those on Earth. I wonder what's the intention of this comparison – does it imply that the manufacturing of the PEC devices could use materials obtained on Moon or Mars within the timeframe of relevance? Would this be necessary (if only say a dozen of PEC devices are needed, so supply on Earth would not be a constraint) or realistic (if instead a high demand is to be met, which would need significant mining/transport)?"*

We appreciate the comment. As part of NASA's and ESA's *in-situ resource utilisation* incentives [[2](https://www.globalspaceexploration.org/wordpress/wp-content/uploads/2021/04/ISECG-ISRU-

20 mA/cm²)’ and STH of 17.9% is contradictory, as 20 mA/cm² is more than 20% STH. The catalysts for ‘realistic exchange current density’ are the ones mentioned in the SI? Please elaborate a bit more here in the main manuscript.”*

We thank the reviewer for the suggestion and have re-written the specified paragraph for increased clarity. The yearly production yield was calculated through scaling the integrated power density of each spectrum (i.e. AM 0 for the Moon) with the integrated power density cycles (for each 1/24 of a lunar month) given in **SI Figure 1**. Then summing up, the total product over each 1/24 of the lunar year is shown. We apologise for the confusion using the example of 20 mA/cm² and 17.9% STH, the reviewer is correct in stating this would be wrong for a terrestrial device (1000 W m⁻²). This was intended, however, for a lunar tandem device experiencing 1367 W m⁻²:

$$\frac{1.23 \text{ V} \cdot 19.85 \text{ mA cm}^{-2} \cdot 10}{1367 \text{ W m}^{-2}} \cdot 100\% = 17.9\% \text{ STH}$$

The catalytic exchange current densities for the ideal case were assumed to be infinite. The statement ‘with the incorporation of realistic catalytic exchange current densities...’ was supposed to signify that realistic, non-infinite values caused a large change impact in the perceived annual production yields. This sentence has been amended in the revised manuscript.

9. *“The Authors go into details of an “Engineering outlook”, but for a space mission, the carry-along-mass of the system is certainly also an important aspect, which should be discussed. See for instance a similar article on PV-powered chemical conversion on Mars [Abel et al, Front. Astron. Space Sci. 9, 2022]. I would guess a PEC system will be heavier than a PV-electrolysis system?”*

This is a valuable comment made by the reviewer. We have carefully studied the reference, could however not find a detailed technical justification for the assumed weight of the PEC device. As stated above as a reply to Reviewer #1’s first comment, a detailed technoeconomic analysis is required in order to compare the two systems with respect to performance and weight/ volume in greater detail. This includes the modelling of interfacial processes and the inclusion state-of-the-

Department of Chemistry
University of Warwick
Gibbet Hill Road
Coventry CV4 7AL

art electrolyser and PEC device designs. This is part of our ongoing research activities which we will hopefully be able to communicate in a separate article in the near future.

10. *“A potentially noteworthy literature context might be that natural, dual-junction PEC water splitting might have taken place on Mars: DOI:10.2138/am.2012.4211.”*

This is a very interesting comment made by the reviewer which is very much appreciated.

Reviewer #3

1. *“What are the noteworthy results?”*

The manuscript presents some noteworthy results on how one could produce oxygen on Mars by splitting CO₂ and on the Moon by splitting H₂O. The idea is not new, but the analysis is.

- *Will the work be of significance to the field and related fields?*

Yes, marginally. The comparison of the performance to that on earth is interesting. However, it is not evident that potential temperature effects were realistically evaluated.”

We would like to thank the reviewer for the comment and would like to point to our response to Reviewer #2, Q4.

2. *“• How does it compare to the established literature? If the work is not original, please provide relevant references.*

As oxygen production on Mars has been demonstrated using high-temperature electrolysis before this work could be published it would be important to first acknowledge that work and second show why the approach here is preferable to what has already been demonstrated. That technology is not even mentioned.”

We thank the reviewer for the valuable comment and added a state-of-the-art oxygen production technology comparison to our ‘Engineering Outlook’.

3. *“• Does the work support the conclusions and claims, or is additional evidence needed? The manuscript seems to recognize the dust in the Martian atmosphere in the atmospheric modeling but not the implication of how to keep dust from settling on the device and degrading its performance. Neither does the manuscript recognize what would happen in a Martian dust storm, which could eliminate production for weeks.”*

Department of Chemistry
University of Warwick
Gibbet Hill Road
Coventry CV4 7AL

This is an interesting and valuable point made by the reviewer. We have since conducted extensive further 2D COMSOL wave optic studies to investigate the relationship between dust deposition layers and transmission as a function of wavelength. We have further contextualised this by providing yearly variance in Martian regolith dust deposition rates to determine how quickly these layers more form. As the reviewer comment indicates, there is a large variance in light transmission and subsequently performance impact during periods where dust storms are more prevalent. This analysis has also been extended to lunar regolith dust and has been added as a separate section to the Results part. After personal correspondence with ESA/ESTEC we decided however that discussing solutions to tackle dust deposition is outside the scope of this work and is a challenge faced by all space engineering, technology and architecture departments.

4. “• Are there any flaws in the data analysis, interpretation, and conclusions? Do these prohibit its publication or require revision?”

Addressing a comparison to state-of-the-art technology does require revision. Addressing dust and dust storms on the performance, not just the atmospheric modeling must also be addressed before considering publication.”

We thank the reviewer for this valuable comment. The questions regarding temperature effects are addressed in our reply to Reviewer #2, Q4, the ones concerning the state-of-the-art technology are addressed in a reply to Reviewer #3, Q2, and an answer regarding the analysis of the potential impact of lunar and Martian dust on device performance is given in a reply to Reviewer #3, Q3. We also include a discussion on state-of-the-art competing O₂ technology as mentioned above.

REVIEWERS' COMMENTS

Reviewer #1 (Remarks to the Author):

I thank the authors for their helpful responses. There are a few residual points for them to consider further, all with reference to my original comments.

Comment 4: This comment and a few from other reviewers were questioning the relationship between the conditions assumed in the modelling and those on Moon or Mars. Where the assumed conditions (e.g. temperature) are different from the ambient conditions, please add a note (e.g. to point out that the assumed working environment of the device is different from its ambient environment).

Comment 5: I suggest that, where you mention the 33mM solubility figure, you add a note of the corresponding T and pCO₂, to avoid misleading the reader.

Comment 7: I am sorry but there seems to be still some confusion with the symbols. In the main text, you introduce U_{theta} (the new equation 5). Is it the same as U_{theta_EPC} in equation S19? If they are identical, then the statement ($V_{op}(i_{op}) = U_{theta}$) (mentioned in both main text, above Eqn. 8, and in SI, above eqn. S19) and eqn. S19 cannot be both correct. If they are different, please explain the difference. Also, it is not entirely clear exactly how i_{op} is calculated – can you provide an equation (or describe a procedure) (in either main text or SI) to show directly how it is done?

Reviewer #2 (Remarks to the Author):

The Authors have amended their manuscript, clarifying many of the points raised by the referees and hereby improving the overall clarity of their work. Yet some new issues were introduced, for instance with an erroneous figure 1, which is why I recommend a major revision.

- It is a very unfortunate style by the Authors to imprecisely describe their changes to the manuscript in their rebuttal and at the same time not providing a document with highlighted changes. This makes the task of the referees unnecessarily tedious.

- In their answers to the referees' comments, the Authors cite literature which then, however, do not appear in the the document's references. As they seem to need these for their reasoning, those references should accordingly be cited in the main document. Furthermore, the data source for their AM spectra is missing.

- The (Quasi-)Fermi levels (QFL) in Figure 1 are still wrong. From their description 'p-type' and 'n-type' material, it looks like they want to describe an idealised Schottky-type of solid-liquid interface without charge-selective layer and homogeneous doping in the respective sub-cell of the tandem, where the top cell is p-type and the bottom cell is n-type. The depicted QFL, on the other hand, shows the situation of a p-n junction, with the QFL recombining near-surface to n-type on the top cell and p-type on the bottom cell. Furthermore, the Fermi-level of the back contact is arbitrarily aligned. For work on how to align these levels, see for instance the book of Peter Würfel on photovoltaics or more recent work with transfer to PEC of Schleuning et al. (DOI:10.1039/D2SE00562J).

- The last sentence in the in the conclusion is quite odd "... harsh environments such as the terrestrial polar regions[67-69]", as none of the three references mentions these polar regions, whereas the reference mentioned in the answer to the comment of Referee 1 (Kolbach et al, Energy Environ. Sci. 14, 2021) does. I guess the authors do not want to claim the idea of transferability to polar regions and simply mixed up references?

Response to reviewer comments

Reviewer #1

1. *"I thank the authors for their helpful responses. There are a few residual points for them to consider further, all with reference to my original comments. Comment 4: This comment and a few from other reviewers were questioning the relationship between the conditions assumed in the modelling and those on Moon or Mars. Where the assumed conditions (e.g. temperature) are different from the ambient conditions, please add a note (e.g. to point out that the assumed working environment of the device is different from its ambient environment)."*

We thank the reviewer for the comment and have noted in the main article the operating environment of the device simulations (unless stated otherwise, they are assumed to follow standard conditions). Moreover, the conditions assumed are justified within the relevant modelling sections.

2. *"Comment 5: I suggest that, where you mention the 33mM solubility figure, you add a note of the corresponding T and pCO₂, to avoid misleading the reader."*

We thank the reviewer for the comment and have amended the article to emphasise that the 33 mM value is quoted "under standard conditions of 298.15 K and 1 atm".

3. *"Comment 7: I am sorry but there seems to be still some confusion with the symbols. In the main text, you introduce U_{theta} (the new equation 5). Is it the same as U_{theta_EPC} in equation S19? If they are identical, then the statement (V_{op}(i_{op}) = U_{theta}) (mentioned in both main text, above Eqn. 8, and in SI, above eqn. S19) and eqn. S19 cannot be both correct. If they are different, please explain the difference."*

We thank the reviewer and apologise for the confusion. U_{theta} represents the thermodynamically required electrochemical potential, i.e. 1.229 V for water-splitting. U_{theta_PEC} represents the actual required electrochemical potential (due to activation overpotentials and resistances), i.e. > 1.229 V. This is the potential that the device will actually need to split water. We have amended SI equation 19 for better clarity and tweaked notation for consistency.

$$V_{op}(i_{op}) = U_{\theta_PEC} = U_{\theta} + |\eta_{-}(\text{cat}, a)| + |\eta_{-}(\text{cat}, c)| + iR_{series}$$

Department of Chemistry
University of Warwick
Gibbet Hill Road
Coventry CV4 7AL

4. *“Also, it is not entirely clear exactly how i_{op} is calculated – can you provide an equation (or describe a procedure) (in either main text or SI) to show directly how it is done?”*

We describe the operating current calculations for water-splitting through SI equation 19, where the operating voltage (V_{op}) is calculated. The corresponding operating current is then calculated through SI equation 13. This has now been highlighted explicitly in the SI for further clarity. “Given a known operating voltage (V_{op} (i_{op})), the corresponding operating current can be calculated with SI equation (13).”.

We describe operating current calculations for CO₂ reduction within the text above SI Figure 6: “Through the previously outlined eq. (1) - (13) in the PEC modelling section, we generate the PV output curve and then solve for the point of intersection between the PV and electrochemical load curves to obtain the device operating current using polyxpoly (MATLAB).” Examples are then given in SI Figure 6, where the operating current is marked by the intersection of the load and PV curves.

Reviewer #2

1. *“The Authors have amended their manuscript, clarifying many of the points raised by the referees and hereby improving the overall clarity of their work. Yet some new issues were introduced, for instance with an erroneous figure 1, which is why I recommend a major revision. It is a very unfortunate style by the Authors to imprecisely describe their changes to the manuscript in their rebuttal and at the same time not providing a document with highlighted changes. This makes the task of the referees unnecessarily tedious.”*

We apologise for not providing a manuscript version highlighting the undertaken changes. With this resubmission we have provided a manuscript version where changes from the previous version are highlighted.

2. *“In their answers to the referees' comments, the Authors cite literature which then, however, do not appear in the the document's references. As they seem to need these for their reasoning, those references should accordingly be cited in the main document. Furthermore, the data source for their AM spectra is missing.”*

We appreciate the comment and have amended the main article to include all references cited in the previous reviewer comment rebuttal and added the data source for the AM spectra.

Department of Chemistry
University of Warwick
Gibbet Hill Road
Coventry CV4 7AL

3. *“The (Quasi-)Fermi levels (QFL) in Figure 1 are still wrong. From their description ‘p-type’ and ‘n-type’ material, it looks like they want to describe an idealised Schottky-type of solid-liquid interface without charge-selective layer and homogeneous doping in the respective sub-cell of the tandem, where the top cell is p-type and the bottom cell is n-type. The depicted QFL, on the other hand, shows the situation of a p-n junction, with the QFL recombining near-surface to n-type on the top cell and p-type on the bottom cell. Furthermore, the Fermi-level of the back contact is arbitrarily aligned. For work on how to align these levels, see for instance the book of Peter Würfel on photovoltaics or more recent work with transfer to PEC of Schleuning et al. (DOI:10.1039/D2SE00562J).”*

We apologise for the mistakes and included a revised version of Figure 1 with the revised manuscript.

4. *“The last sentence in the in the conclusion is quite odd “... harsh environments such as the terrestrial polar regions[67-69]”, as none of the three references mentions these polar regions, whereas the reference mentioned in the answer to the comment of Referee 1 (Kolbach et al, Energy Environ. Sci. 14, 2021) does. I guess the authors do not want to claim the idea of transferability to polar regions and simply mixed up references?”*

The article quote in the question “Moreover, it opens the possibility of exploring (photo-) electrochemical devices as well in other harsh environments such as the terrestrial polar regions⁶⁷⁻⁶⁹”. Cites three references.

67 - “Sargeant, E. et al. Electrochemical Conversion of CO₂ and CH₄ at Subzero Temperatures. *ACS Catal.* **10**, 7464-7474 (2020).”

68 - “Rabinowitz, J. A. & Kanan, M. W. The future of low-temperature carbon dioxide electrolysis depends on solving one basic problem. *Nat. Commun.* **11**, 5231 (2020).”

69 - “Küngas, R. Review-Electrochemical CO₂ Reduction for CO Production: Comparison of Low- and High-Temperature Electrolysis Technologies. *J. Electrochem. Soc.* **167**, 044508 (2020).”

Each reference explores electrochemical CO₂ reduction at low temperatures. The references support the statement that there could be the “possibility” to explore PEC devices in harsh environments because CO₂ reduction has already been demonstrated in temperature conditions below 273.15 K which is transferable to cold polar regions. Furthermore reference 67 explicitly states in the introduction that “These reactions can then be used as a sustainable energy cycle, in a reversible fuel cell, **under severe environmental conditions.**” We apologise

Department of Chemistry
University of Warwick
Gibbet Hill Road
Coventry CV4 7AL

however for the potential confusion that these references refer to CO₂ reduction investigations in polar regions and added the suggested citation.